# BTR: Binary Token Representations for Efficient Retrieval-Augmented Language Models

**Qingqing Cao, Sewon Min, Yizhong Wang, Hannaneh Hajishirzi**
Paul G. Allen School of Computer Science & Engineering
University of Washington
`{qicao,sewon,yizhongw,hannaneh}@cs.washington.edu`

## Abstract

Retrieval augmentation addresses many critical problems in large language models such as hallucination, staleness, and privacy leaks. However, running retrieval-augmented language models (LMs) is slow and difficult to scale due to processing large amounts of retrieved text. We introduce binary token representations (BTR), which use 1-bit vectors to precompute every token in passages, significantly reducing computation during inference. Despite the potential loss of accuracy, our new calibration techniques and training objectives restore performance. Combined with offline and runtime compression, this only requires 127GB of disk space for encoding 3 billion tokens in Wikipedia. Our experiments show that on five knowledge-intensive NLP tasks, BTR accelerates state-of-the-art retrieval-augmented language model inference by up to 4x and reduces storage by over 100x while maintaining over 95% task performance. [1]

## 1 Introduction

Large language models (LLMs) (Brown et al., 2020b; Touvron et al., 2023a), despite their widespread success, still suffer from issues such as hallucinations (Mallen et al., 2023; Mündler et al., 2023), staleness, and privacy leaks (Huang et al., 2022). Retrieval-augmented language models (e.g., (Lewis et al., 2020; Izacard et al., 2022b)) alleviate such problems via a retrieve-and-read approach (Figure 1), using a retrieval component to find passages relevant to the input query followed by a reader model (e.g., an LLM) that generates output given the query concatenated with retrieved passages. However, this approach is slow (e.g., only handles a few queries for a powerful server GPU) at inference time mainly due to the reader model that needs to compute cross-attention between the input query and a large number of passages. Existing solutions (Lee et al., 2021; Cao et al., 2020) improve the inference computation by decomposing query and passage encoding and precomputing passage representations, but they come with large storage costs (terabytes of storage; §4).

In this work, we improve the inference speed of the reader model in a retrieval-augmented LM with a small storage footprint by introducing *cacheable and calibrated **B**inary **T**oken **R**epresentations* (**BTR**). BTR (Figure 1) precomputes token representations for the retrieved passages in the reader model. The binary representations are 1-bit vectors for the tokens in a passage and are created from the hidden states in the reader encoder layers via *calibrated binarization* which are effective for downstream tasks such as question answering. BTR reduces the storage footprint and improves the runtime speed since the representations are 1-bit vectors, and the reader uses the cached representations. To avoid degradation of task accuracy caused by binary representations, we introduce two training objectives by adding (i) a *passage representation recovery* objective that makes the binary representations preserve the passage semantics before the binarization; and (ii) a *query-aware passage token distillation* objective that compensates the information loss due to precompuation of passage representations independent of the query.

Furthermore, we observe significant redundancies in precomputed token representations in retrieved passages since they are relevant to the query and contain similar information. Removing such redundancies only causes minimal task accuracy loss (<0.5% in our experiments), but shows significant

---

[1] Our code is publicly available at `https://github.com/csarron/BTR`

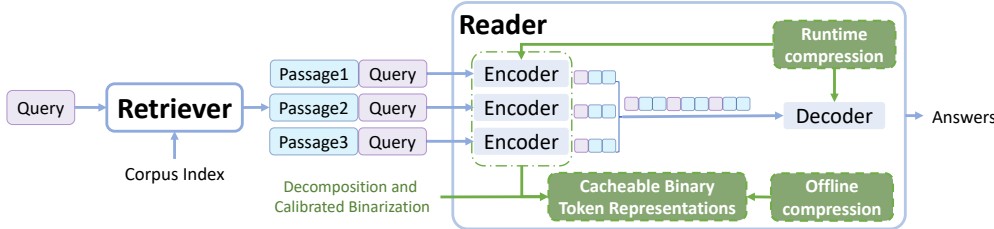

Figure 1: Retrieval-augmented models use a retrieve-and-read pipeline. The reader can be either an encoder or an encoder-decoder model. BTR creates cacheable binary representations for the passages via decomposition and calibrated binarization to speed up reader inference. BTR further reduces storage by offline compression and improves inference speed by runtime compression.

benefits in storage and runtime inference. To address this, we further develop token compression over BTR by merging similar precomputed token vectors after training and merging similar concatenated query-passage representations during inference.

We evaluate BTR on five knowledge-rich NLP tasks, including three open-domain question-answering tasks (NaturalQuestions (Kwiatkowski et al., 2019), TriviaQA (Joshi et al., 2017), and WebQA (Berant et al., 2013)), the FEVER (Thorne et al., 2018) fact-checking task, and the MMLU (Hendrycks et al., 2020) reasoning benchmark. Compared to baseline systems, BTR reduces disk storage by up to **101x** and improves inference speed by **2–4x** for the reader of two state-of-the-art retrieval-augmented language models. BTR also retains 90–95% of the original models' performance. Our analysis experiments show that binary token representations are effective and contribute most to improving the inference speed and reducing the storage costs of readers. The training regularization objectives help mitigate the task accuracy loss and the offline and runtime token compression techniques further bring down storage footprint and increase inference efficiency.

## 2 BACKGROUND AND RELATED WORK

We first describe the architecture of retrieval-augmented language models and then related efficiency techniques designed for them. Next, we summarize relevant methods to improve model efficiency, including quantization and binary representations.

**Retrieval-Augmented Language Models.** Retrieval-augmented language models have shown strong performance in many tasks, including language modeling (Borgeaud et al., 2022; Min et al., 2023), open-domain question answering, and fact checking (Izacard et al., 2022b; Lewis et al., 2020; Guu et al., 2020; Shi et al., 2023).

As shown in Figure 1, a retrieval-augmented language model works by first using a retriever to retrieve many passages relevant to the input query and then using a reader component to extract or generate the answers. While the retriever is often fast enough, the reader causes a speed bottleneck in retrieval augmentation because it requires computing cross-attention between the input query and many passages. For example, Figure 1 illustrates a state-of-the-art encoder-decoder reader (Izacard et al., 2022b) architecture called Fusion-in-Decoder (FiD) (Izacard & Grave, 2021). FiD first concatenates each passage with the query and processes them in parallel (independently) in the encoder; then the decoder fuses information across all the concatenated passage-query representations and produces answers. In our experiments, we observe that the passage encoding takes over 60% of the reader's computation on commodity GPUs, and we save such computations by precomputing the passage representations, leading to significant speedup in the inference with reduced storage costs.

**Efficient methods for reader models.** DensePhrase (Lee et al., 2021) builds contextualized phrase representations for the passage corpus, completely removes the reader component, and uses phrase retrieval to produce answers for the query. Despite its high inference throughput, the accuracy is much lower than similar size BTR models. FiD-light (Hofstätter et al., 2022) and FiDO (de Jong et al., 2022) focus on improving the inference latency for FiD models on customized hardware like TPUs, they compress passage representations into fixed-size vectors to reduce the decoder computation.

But on more popular hardware like GPUs, the passage encoder is the computation bottleneck, and their decoder-oriented optimizations are likely to be less effective in improving inference speed. LUMEN (de Jong et al., 2023) and DeFormer (Cao et al., 2020) precompute cacheable continuous passage representations to speed up reader inference, but they take up 100x more storage than BTR.

**Model quantization and binarization.** Another line of research focuses on model quantization (Dettmers et al., 2022; Xiao et al., 2023; Yuan et al., 2023; Frantar et al., 2022) or binarization (Bai et al., 2021; Qin et al., 2022) for resource-constrained settings. It improves model efficiency by compressing model weights to lower precision to reduce runtime memory usage. However, quantization does not necessarily improve the inference speed of readers as we will empirically show in §4. This is because quantized models still need to process large amounts of input passage data for readers, which requires huge computation. Meanwhile, quantization methods are orthogonal to our BTR and can help deploy BTR for platforms with low resources.

**Binary representations.** Previous work has used binary representations for text retrieval (Yamada et al., 2021), semantic matching (Tissier et al., 2019; Shen et al., 2019), or image retrieval (Jain et al., 2017; Huang et al., 2019; Liu et al., 2016). Motivated by this line of work, we build binary token-level passage representations to improve reader efficiency which was unexplored before. The key difference is that existing passage-level (or image-level) binary representations provide coarse-grained semantic signals that work for retrieval or semantic matching. But token-level representations require storing more fine-grained information, which we effectively address by introducing new calibration techniques and training objectives.

## 3 METHOD

We present BTR, a fast and storage-efficient reader for retrieval-augmented language models. BTR creates cacheable calibrated binary representations for the retrieved passages to speed up inference and avoid large storage overhead. We describe BTR for encoder-decoder readers which are most widely used in the state-of-the-art models (Izacard et al., 2022b; Izacard & Grave, 2021), and later show how to apply BTR for the encoder-only model (§4). The encoder and encoder-decoder readers both need to process many passages, which is the computation bottleneck. Previous solutions (Cao et al., 2020; Lee et al., 2021) precompute the passage representations to avoid runtime computation but come with large amounts of storage costs (e.g. terabytes of storage for Wikipedia scale corpus). BTR tackles the challenge by building compact binary token representations for the passages.

**Overview.** Figure 2 summarizes the BTR reader architecture and how our techniques are applied to each component. The key idea in BTR is to create **cacheable binary token representations** (§3.1) of the retrieved passages such that the passage encoding can be precomputed offline and stored in a compact format. We build on the previous technique (Cao et al., 2020) to decompose the passage and query computation in the lower layers of the reader encoder and jointly process the query and passage representations back in the upper layers. However, unlike them, we create calibrated binary passage representations after the decomposition to drastically reduce storage. We further develop **offline compression** to reduce the storage of these precomputed binary token representations. Such binarization and decomposition incur task performance degradation, which we address by designing two regularization techniques during training (§3.2). For inference, we develop **runtime compression** techniques (§3.3) to further speed up the reader computation.

### 3.1 BINARY TOKEN REPRESENTATIONS FOR RETRIEVED PASSAGES

Our goal is to produce binary representations for the retrieved passages while maintaining downstream task accuracy. We binarize passage representations at the token level. Specifically, given a continuous passage token vector $\mathbf{h}_k = [h_1, h_2, \cdots, h_d]$ ($d$ is dimension) at the $k$th layer of the reader encoder, we hash $\mathbf{h}_k$ via the $\mathrm{sign}$ function to obtain the binary representation vector $\mathbf{b}_k = \mathrm{sign}(\mathbf{h}_k)$, where $b_i$ is 1 if $h_i > 0$ and $-1$ otherwise. We use a differentiable $\tanh$ function to approximate the non-differentiable sign function (Cao et al., 2017).

Prior work (Yamada et al., 2021) binarized passage representations for retrieval tasks through an annealing process, i.e., $\tilde{\mathbf{b}} = \tanh(\beta\mathbf{h})$, where $\beta$ is a scaling factor that anneals to infinity to

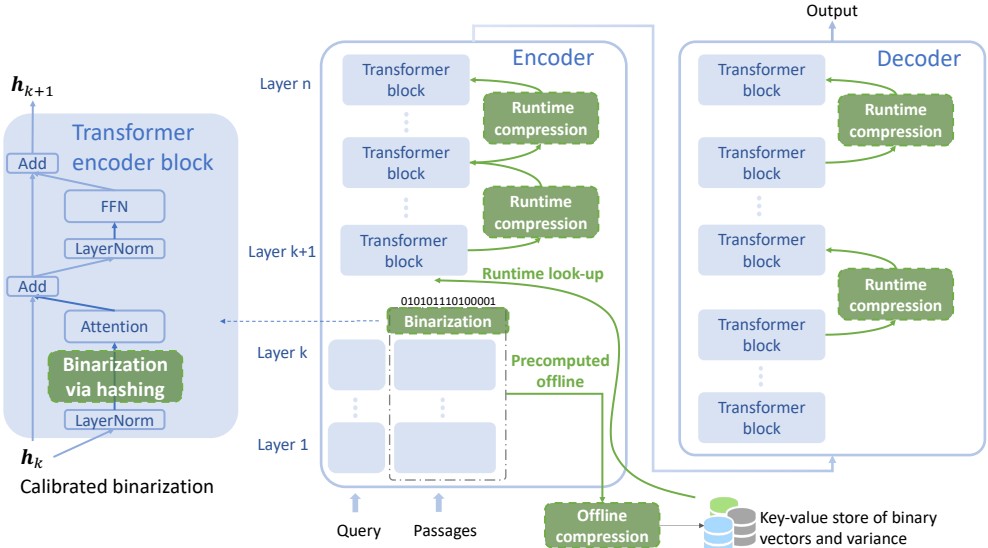

Figure 2: BTR reader architecture, where light blue color indicates the model from prior work (based on T5 (Raffel et al., 2020) and FiD (Izacard & Grave, 2021), and green indicates our methods. We create cacheable calibrated binary token representations for retrieved passages in the reader encoder to speed up inference. Additionally, we compress the precomputed binary token presentations offline to reduce storage costs. We further reduce inference computation by runtime compression for the encoder and decoder.

approximate the sign function. However, we find that directly converting the continuous float values into discrete binary values with annealing decreases the representation quality. The problem is that the float vector values often have large scales (e.g., ranging from -500 to +300) which causes the tanh function to collapse into -1 and +1 quickly, this is partially due to the pre-layernorm (Xiong et al., 2020) nature of the encoder-decoder architectures like T5. Most recent models like GPT-3 (Brown et al., 2020a) and LLaMA (Touvron et al., 2023b;a) also use pre-layernorm. A naive fix is to introduce a normalization layer (e.g. layernorm) before the binarization, but then the scale of normalized values does not match the input passage presentations in the upper encoder layers and we empirically notice a larger drop in performance.

**Calibrated binarization.** We introduce a calibration technique that makes two modifications to the pre-layernorm architecture.[2] First, instead of applying the binarization after the Transformer encoder layer as in the previous work (Shen et al., 2019; Yamada et al., 2021), we insert the binarization after the layernorm layer (but before the multihead attention layer) inside the Transformer encoder layer as shown in Figure 2. We also save the variance of all passage token representations which comes with negligible storage costs[3]. Using the saved variance information along with the layernorm weights, we could recover the passage representations to their original scales (by dividing the product of variance and layernorm weights). Second, during training, instead of annealing the tanh function, we adopt a straight-through estimatior (Bengio et al., 2013) for the binarization, the forward pass of encoding still uses discrete binary values while the backward propagation uses the tanh function to provide the gradients. These two key changes make the binarization procedure better calibrated and help retain the representation quality after binarization.

**Offline compression.** The binary representations for the passage tokens are amenable to further compression due to *context redundancy*. This redundancy originates from the passage corpus, where each token can occur millions of times in different contexts even though many of them have similar semantic information. This is especially the case for tokens that form stopwords. We design offline token compression that aims to reduce such redundancy such that we do not need to store its representations every time it appears in a different context (in our case, the context is a retrieved

---

[2]Not necessarily needed for models with post-layernorm such as BERT.

[3]It requires $1/d$ storage of the passage vector. $d$ is usually several hundred or thousand like 768 or 1024.

passage). For a given token $t$ in the reader model's vocabulary, we find all binary representations $T$ from the corpus. If the token $t$ is a stopword $t_s$, then we compute the mean vector values $\mathbf{b}_s$ for all the tokens in $T$. Otherwise, we merge $r_o\%^4$ of the binary token representations $\mathbf{b}_v$ for the token $t_v$ based on their semantic similarity. We use a bipartite merging (Bolya et al., 2022) algorithm and measure the semantic similarity using the Hamming distance. We find this more efficient than a clustering method as they require more iterations and do not easily scale to billions of tokens. Details are provided in Algorithm 1 in Appendix A.1. Once we compress the binary token representations, we save them in a key-value data store for fast runtime lookup.

## 3.2 BTR TRAINING

Binarization and decomposition provide efficiency benefits but come with performance degradation for downstream tasks. We describe how we train BTR in three steps to mitigate accuracy drops.

- **Step 1.** We first train a reader model without any decomposition (precomputation) or binarization. The training objective is the task loss $\mathcal{L}_{\text{task}}$ using cross entropy between prediction logits and task labels.

- **Step 2.** We then train a decomposed (no binarization) model with the reader model from step 1 as the teacher using *query-aware passage token distillation* loss $\mathcal{L}_{\text{distill}}$ and the original task loss $\mathcal{L}_{\text{task}}$ (described below).

- **Step 3.** Finally, we train a decomposed reader model by applying BTR binarization in the reader encoder, where the model weights are initialized from the distilled decomposed reader model in the second step. We add a *passage representation recovery* loss $\mathcal{L}_{recovery}$ to the original task loss $\mathcal{L}_{\text{task}}$ (described below).

**Query-Aware passage token distillation.** Decomposing passages encoding from the query in the lower layers of the reader encoder often causes task performance degradation. Previous work (Cao et al., 2020) distilled upper-layer passage representations in the decomposed reader encoder from the original (not decomposed) model to reduce information loss due to decomposition. We empirically find that the direct application of this distillation does not improve the task performance much for a retrieval-augmented LM. This is likely because DeFormer-style distillation applies to single passage scenarios where all passage tokens are important to the task, whereas in our setting the reader retrieves many passages that may contain more redundant passage token information. Distilling all token representations using Cao et al. (2020) from all retrieved passages distracts the reader encoder and makes it fail to pass useful passage token information for the task.

To address this issue, we design a query-aware passage token distillation loss to improve the decomposed passage representation quality. We only distill the passage tokens that are semantically relevant (salient) to the query from the original (non-decomposed) model. This is because query-relevant passage tokens carry more fine-grained query-related information from the original (not decomposed) passage representations. We obtain top-$r^5$ passage tokens using the query to passage attention scores. $\mathcal{L}_{\text{distill}} = \frac{1}{r}\sum_{i=1}^{r}(h_i - h_i^{\text{decomposed}})^2$, where $h_i^{\text{decomposed}}$ is the decomposed passage representations. The second step training loss is: $\mathcal{L}_{\text{task}} + \mathcal{L}_{\text{distill}}$.

**Passage representation recovery.** The end-to-end training objectives do not explicitly encourage the binary tokens representations to maintain semantic information in the continuous representations before binarization. We add a representation recovery loss (Shen et al., 2019) to the second-stage training task objective. This objective directly provides supervision signals for the binary representations to retain the semantics of continuous representations. Specifically, we first add a linear projection layer (parameterized by $\mathbf{W}_{proj}$ and $\mathbf{bias}_{proj}$) on top of the binary passage representations $\mathbf{b}_k$; then we use the mean square error as the recovery loss between the projected representations and original passage representations $\mathbf{h}_k$ before the binarization ($k$th layer in the reader encoder): $\mathcal{L}_{recovery} = \frac{1}{d}\sum_{i=1}^{d}(h_i - b_i^{proj})^2$, where $\mathbf{b}^{proj} = \mathbf{W}_{proj}\mathbf{b} + \mathbf{bias}_{proj}$. As we will show in §4.4, passage recovery loss helps improve the quality of binary representations and reduce the accuracy gap. The third step training loss is $\mathcal{L}_{\text{task}} + \mathcal{L}_{recovery}$.

---

[4] $r_o\%$ is 100% for stopword token representation since stopwords are less contextual than other tokens.

[5] We set $r$ to 50% of total passage tokens by default.

### 3.3 BTR INFERENCE

During inference, given an input query and retrieved passages, BTR reader first computes query representations (continuous float-point vectors) on the fly in the encoder (layer 1 to $k$), then it looks up the precomputed binary token representations (from layer $k$) for the retrieved passages from the data store. Next, BTR reader converts the loaded binary token representations into continuous float-point vectors[6] and concatenates them with query representations before layer $k + 1$. Then from layer $k + 1$ to layer $n$ in the reader encoder, BTR applies *runtime compression* (described below) over the concatenated query-passage representations to reduce the number of tokens and further speed up the encoder inference. Once the encoder finishes computation, the reader flattens and concatenates all the query-passage representations, and then feeds them into the reader decoder. The reader decoder applies similar runtime compression operations to shorten the concatenated text sequence with query and all passage tokens.

**Runtime compression.** Runtime compression is possible because the retrieved passages are relevant to the query and contain similar information for different passages. The runtime token compression consists of intra-passage and cross-passage compression steps. At the upper layers ($k + 1$ to $n$) in the reader encoder, the query representations are concatenated with representations of each passage, and the query-passage representations are processed together. We compress the token representations of each query-passage pair and call this process intra-passage token compression since the encoder still computes each query-passage pair representation in parallel. After the encoder computation, all the query-passage representations are flattened and concatenated. In the reader decoder, we further compress the resulting representations across all passages (and the query) for every $g$ layer[7] and call this cross-passage token compression. We use a similar compression algorithm as in the offline token compression, merging $r_p\%$ tokens based on their semantic similarity. The difference here is that the representations are continuous vectors in the upper encoder and the decoder, so we change the similarity metric to cosine distance accordingly. Details are in Algorithm 2 in Appendix A.1.

## 4 EVALUATION

We first describe BTR and baselines, followed by five knowledge-rich NLP tasks to evaluate them. We then report the main results of BTR on task performance and efficiency gains. We also evaluate the proposed techniques in BTR and their impacts on efficiency and performance trade-offs.

### 4.1 BASELINES

We apply BTR to Atlas (Izacard et al., 2022b) base and large variants and call them BTR-Atlas base and BTR-Atlas large. Atlas is a recent state-of-the-art retrieval-augmented language model. The Atlas reader uses the T5-based (Raffel et al., 2020) Fusion-in-Decoder (FiD) (Izacard & Grave, 2021) architecture. We include more implementation details in Appendix A.2.

**Baselines.** To compare task performance and efficiency with BTR, we evaluate five representative baselines (implementation details are in Appendix A.3):

(1) **Atlas** (Izacard et al., 2022b) is a state-of-the-art retrieval-augmented language model for knowledge-rich NLP tasks. Due to computing budget constraints, we compare the base (with 220M parameters) and large (770M parameters) size variants.

(2) **Atlas-Q** is a quantized version of the Atlas base model. We use GTPQ (Frantar et al., 2023) to quantize the fine-tuned Atlas baseline model in 4bit.

(3) **DensePhrase** (Lee et al., 2021) creates dense phrase representations of passages. It retrieves answers from the phrase index without using the retrieve-and-read pipeline and achieves high query processing throughput.

(4) **DeFormer** (Cao et al., 2020) speeds up the inference in encoder-only reader models by decomposing the passage and query encoding and caching the continuous passage representations. We apply DeFormer over Atlas-base and fine-tune the decomposed model.

---

[6]values are -1.0 and 1.0.)

[7]We set $g$ to be 3 as default. We do not merge in every layer as in the upper part of the encoder because the decoder has more layers, skipping compression for a few layers provides a balance between efficiency and accuracy.

Table 1: Task performance and inference efficiency comparison between the baseline models and BTR-Atlas over five evaluation tasks. 'Sp' denotes the speed up over the Atlas base model, and the subscript percentage numbers in brackets are relative performance compared to the Atlas base model. The speed up and percentage for BTR-Atlas large is over the Atlas large model. BTR achieves **2–3x** inference speedup while maintaining $> \mathbf{95\%}$ of the original models' performance.

| | NQ | | TQA | | WQ | | FEVER | | MMLU | |
|---|---|---|---|---|---|---|---|---|---|---|
| | EM | Sp | EM | Sp | EM | Sp | Acc | Sp | Acc | Sp |
| Atlas base | 52.1 | 1.0 | 69.3 | 1.0 | 46.4 | 1.0 | 72.9 | 1.0 | 38.6 | 1.0 |
| Atlas-Q | $51.8_{(99\%)}$ | 1.2 | $68.5_{(99\%)}$ | 1.3 | $45.1_{(97\%)}$ | 1.1 | $70.4_{(97\%)}$ | 1.1 | $37.8_{(98\%)}$ | 1.2 |
| DeFormer | $51.4_{(99\%)}$ | 2.2 | $68.0_{(98\%)}$ | 2.0 | $44.8_{(97\%)}$ | 2.0 | $71.8_{(98\%)}$ | 2.3 | $33.9_{(88\%)}$ | 1.8 |
| DensePhrase | $40.9_{(79\%)}$ | 4.9 | $53.6_{(77\%)}$ | 5.8 | $37.5_{(81\%)}$ | 5.1 | - | - | - | - |
| LLAMA2-7B | $47.8_{(92\%)}$ | 0.1 | $74.3_{(108\%)}$ | 0.1 | $51.2_{(110\%)}$ | 0.1 | $76.3_{(105\%)}$ | 0.1 | $51.2_{(133\%)}$ | 0.1 |
| **BTR**-Atlas base | $49.5_{(95\%)}$ | 3.1 | $66.7_{(96\%)}$ | 2.5 | $43.8_{(94\%)}$ | 2.6 | $70.2_{(96\%)}$ | 3.1 | $35.4_{(92\%)}$ | 2.6 |
| Atlas large | 58.3 | 1.0 | 73.6 | 1.0 | 51.5 | 1.0 | 78.2 | 1.0 | 41.1 | 1.0 |
| **BTR**-Atlas large | $56.1_{(96\%)}$ | 4.0 | $70.8_{(96\%)}$ | 3.9 | $49.1_{(95\%)}$ | 3.6 | $75.9_{(97\%)}$ | 4.1 | $39.2_{(95\%)}$ | 2.5 |

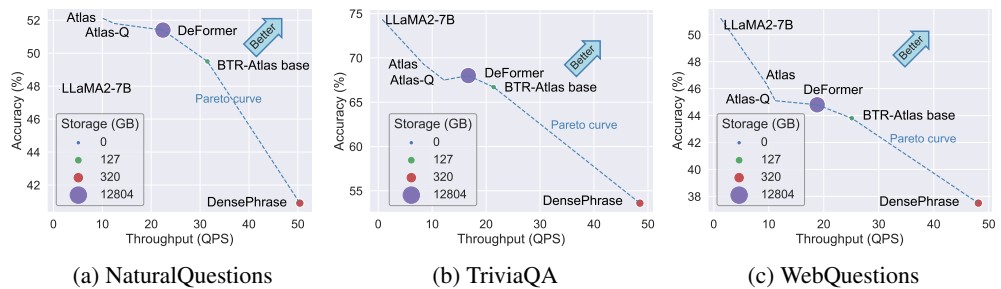

(a) NaturalQuestions        (b) TriviaQA        (c) WebQuestions

Figure 3: Task performance and inference efficiency (throughput and storage) visualization across the baselines and BTR fine-tuned models on three open-domain QA datasets. The area of the circle denotes storage size (we scale the area size of BTR-base and DensePhrase by 10 for clear display). A top-right smaller circle is better.

(5) **LLaMA2** (Touvron et al., 2023b) consists of open-source large language models that achieve top performance on many NLP benchmarks including the knowledge tasks we study. Since the few-shot versions still lag behind fine-tuned models, we finetune the LLaMA2-7B model using the LoRA (Hu et al., 2021) adapter and report its results on the evaluation tasks.

## 4.2 TASKS AND METRICS

We evaluate BTR and baselines on three open-domain QA tasks: NaturalQuestions (NQ, Kwiatkowski et al. (2019), TriviaQA (TQA, Joshi et al. (2017)), WebQuestions (WQ, Berant et al. (2013)); one fact-checking task: FEVER (Thorne et al., 2018), and one knowledge-intensive reasoning benchmark: the mass-multitask language understanding (MMLU) dataset (Hendrycks et al., 2020). Dataset details are in Appendix A.4. We use the same Wikipedia corpus (dumped from December 2018) for all tasks and compared systems for consistency. The corpus contains 32.1 million passages, roughly over 3.2 billion tokens assuming 100 tokens per passage.

**Evaluation metrics.**    We use the exact match score (EM) as the metric for the open-domain QA tasks. We measure the accuracy of the fact checking task and the MMLU benchmark. For all tasks and models, we choose the model checkpoint based on the development set and report the metrics on the test set. All the numbers are average values across 5 different runs (with different random seeds).

**Efficiency measurements.**    For each task and model, we randomly sample 100 queries from the test dataset and measure the real inference throughput as the number of queries processed per second (QPS) of the compared models on the A100 GPU hardware. Actual throughput reflects a more realistic comparison of efficiency across different systems.

Table 2: Inference throughput and accuracy for applying binary representations to BERT-base reader model on three open QA datasets.

| | NQ | | TQA | | WQ | |
|---|---|---|---|---|---|---|
| | EM | Sp | EM | Sp | EM | Sp |
| BERT base | 42.9 | 1.0 | 57.1 | 1.0 | 43.1 | 1.0 |
| **BTR**-BERT base | 39.8 (93%) | 3.4 | 54.1 (95%) | 3.9 | 39.7 (92%) | 3.2 |

Table 3: Ablation analysis for each component in BTR on the NaturalQuestions dataset.

| | | Accuracy (%) ↑ | Throughput (QPS) ↑ | Storage (GB) ↓ |
|---|---|---|---|---|
| | Atlas base | 52.1 | 10.2 | 0 |
| | **BTR**-Atlas base | 49.5 | 31.5 | 127 |
| efficiency | w/o binary passage representations | 50.8 (+1.3) | 32.3 | 12,804 |
| | w/o offline compression | 49.8 (+0.3) | 31.5 | 159 |
| | w/o runtime intra-passage compression | 50.0 (+0.5) | 28.1 | 127 |
| | w/o runtime cross-passage compression | 49.9 (+0.4) | 24.6 | 127 |
| accuracy | w/o passage representation recovery | 47.4 (-2.1) | 31.5 | 127 |
| | w/o query-aware passage token distillation | 48.2 (-1.3) | 31.5 | 127 |

## 4.3    MAIN RESULTS

**BTR achieves favorable accuracy and efficiency tradeoffs.**    Table 1 presents the accuracy, storage versus inference throughput trade-offs for BTR, DeFormer, DensePhrase, Atlas-Q, and LLaMA2-7B models on the evaluation datasets. We define the better efficiency trade offs under certain a accuracy target (e.g, afford relatively 5% accuracy drop).  A model has a better trade-off if the model is more efficient (less storage and faster speed) when satisfying the accuracy budget.  Atlas-Q and LLaMA-7B provide high accuracy but with very low inference throughput. DensePhrase achieves the highest throughput, but comes with much lower accuracy numbers ($> 6\%$ gap); DeFormer has similar inference throughput and slightly higher accuracy than BTR but its storage overhead is two orders of magnitude larger.   Figure 3 visualizes the efficiency versus accuracy trade-offs for the compared models on the three open QA datasets.

Furthermore, BTR is more scalable than DeFormer and DensePhrase. We estimate that using a much larger corpus like CCNet (Wenzek et al., 2020) with over 350 million passages will take up 1389 GB using similar compression ratios for the Wikipedia corpus. However, for baseline models, the DeFormer will spend over 100TB of storage and the DensePhrase will occupy over 3TB of storage.

**BTR remains effective for encoder-only models.**    We study binary passage token representations for encoder-based models and validate their effectiveness and inference benefits. Specifically, we apply the token-level binarization technique in §3.1 for the BERT-base model with DPR (Karpukhin et al., 2020) as the passage retriever. We do not apply scale-aware modifications since the BERT model is a post-layernorm architecture. Note that encoder models expect the answer predictions to be exact spans from the retrieved passages, therefore, our runtime token compression techniques do not apply since we cannot recover the original exact spans once the tokens are compressed. Offline binary token compression remains applicable and reduces storage footprint. Implementation details are in Appendix A.2. Table 2 shows the results of the BERT reader model on three open QA datasets, applying binary representations also effectively improves the inference throughput by over **3x** while maintaining over 92% accuracy.

## 4.4    ABLATION STUDY

**Each component in BTR provides either efficiency or accuracy gains.**    We study the effects of each component in BTR on the inference efficiency and accuracy using the NaturalQuestions dataset. BTR consists of four efficiency components that improve either inference throughput or storage footprint and two training components that improve accuracy. The efficiency components include (i) binary passage representations; (ii) offline token merging; (iii) runtime intra-passage token compression; and (iv) runtime cross-passage token compression. The accuracy components are (v) passage representation recovery and (vi) query-aware passage token distillation.

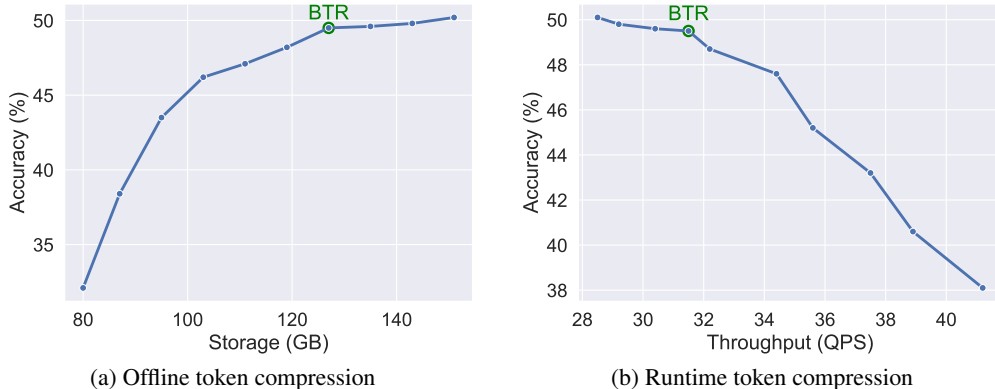

(a) Offline token compression                    (b) Runtime token compression

Figure 4: Accuracy, storage, and throughput comparisons for different two-stage token compression ratios on the NaturalQuestions dataset. To achieve a good balance between accuracy, storage, and throughput in BTR, we choose a compression ratio of 0.2 for both online and offline token compression. Detailed numbers are in Table 9 in Appendix A.6.

Table 3 shows the efficiency techniques in BTR come with the price of task accuracy degradation, however the two accuracy components during training effectively mitigate such accuracy drop. Specifically, the binary passage representations dramatically reduce the storage by over 100x, and the offline token compression further reduces the footprint by 20%. On the other hand, the two online token compression techniques improve the inference throughput by over 30% without sacrificing too much accuracy drop. Overall, the techniques in BTR collectively provide efficiency gains without compromising task accuracy too much.

**Effects of token compression ratios in BTR.**    Figure 4 shows the effects of the two-stage token compression technique in BTR under different compression ratios. We choose the compression ratio from 5% to 50%[8] with a gap of 5% and report the accuracy and storage or throughput numbers for the NaturalQuestions dataset. For offline token compression, with bigger compression ratios, BTR achieves linear storage cost reduction but the accuracy does not drop significantly until the ratio is bigger than 0.2 (default for BTR). We also observe that for online token compression, the inference throughput consistently improves with higher compression ratios, while the accuracy degradation is relatively small until the ratio is larger than 0.2. By configuring different compression ratios, BTR allows flexible inference throughput, storage, and accuracy trade-offs during inference without additional training efforts.

## 5    CONCLUSION AND FUTURE WORK

Retrieval-augmented language models present viable solutions to address issues like hallucinations in large language models, but their inference is slow because the reader component needs to process many passages. We introduce BTR, a system that creates cacheable binary token representations to improve the inference and storage efficiency for the reader in retrieval-augmented language models. Together with training regularization techniques and token compression methods, BTR effectively reduces the storage costs by 100x, improves inference throughput by 2∼4x, and maintains performance for a wide range of knowledge-intensive tasks.

We discuss directions for future work. 1) Extending BTR to decoder-only readers is non-trivia because decoder models compute passage representations together with the query in a sequential manner, making it challenging to break computational dependencies and cache passage representations. Moreover, the KV-Caches in decoder models speed up the inference decoding, but storing their binary representation causes much more storage than encoder models. 2) Improving BTR for extremely long input queries remains challenging and requires other orthogonal efficient methods. BTR can speed up the inference when queries are longer, but the speed-up is relatively smaller than shorter queries. 3) Scaling BTR for larger models with bigger representation sizes is another important research topic. Potential solutions might include using autoencoders to compress the dimension of representations. 4) It will also be interesting to apply binary token representations to the retriever and incorporate BTR into model pretraining for building better and faster retrieval-augmented language models.

---

[8]Compression ratios bigger than 50% cause significant accuracy drops.

ACKNOWLEDGEMENTS

This research was supported partly by NSF IIS-2044660, an Allen Investigator Distinguished award. We thank the members of the UW NLP group for their comments and feedback on this paper.

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

# A APPENDIX

## A.1 TOKEN COMPRESSION ALGORITHM

---

**Algorithm 1** Offline Compression for Binary Token Representations

---

**Input:** precomputed binary passage token representations $\mathcal{B}$ over a corpus $\mathcal{C}$, compression ratio $r_o$, the vocabulary of reader model $\mathcal{V}$
**Output:** compressed binary passage token vectors $\mathcal{B}_c$

1: **for** each stopword token $t_s$ in $\mathcal{V}$, **do**
2:    collect all binary representations $\mathbf{B}_s$ in corpus $\mathcal{C}$
3:    compute the average across all binary representations: $\overline{\mathbf{b}}_s = \text{sign}(\frac{1}{|\mathbf{B}_s|} \sum \mathbf{B}_s)$
4: **end for**
5: **for** each token $t_v$ in $\mathcal{V}$ that is non-stopword, **do**
6:    collect all binary representations $\mathcal{B}_v$ in corpus $\mathcal{C}$
7:    merge $\mathbf{B}_v$ using hamming distance $\mathcal{H}$: $\mathcal{B}_v = \text{sign}(\text{bipartite\_merge}(\mathbf{B}_v, r_o, \mathcal{H}))$;
8: **end for**
9: ──────────────────────────────
10: **function** BIPARTITE_MERGE(input tokens: $\mathbf{B}$, merge ratio: $r$, distance metric $\mathcal{M}$)
11:    divide tokens $\mathbf{B}$ into two sets of tokens $\mathbf{B}_i$ and $\mathbf{B}_j$ based on even and odd order
12:    **for** each token in $\mathbf{B}_i$, **do**
13:       create an token edge $e$ to its most similar token in $\mathbf{B}_j$ using $\mathcal{M}$
14:       store token edge $e$ into $\mathbf{B}$
15:    **end for**                                        ▷ this is implemented as a fast parallel operation
16:    Keep top-$r'$ edges $\mathbf{E}_r$, where $r' = r|\mathbf{X}|$
17:    for each token edge in $\mathbf{E}_r$, merge connected tokens in $\mathbf{B}_i$ and $\mathbf{B}_j$ into $\mathbf{B}_e$ by computing the average of token vectors, gather the rest unmerged tokens $\mathbf{B}_i^{rest}$ and $\mathbf{B}_j^{rest}$ **return** merged tokens $\mathbf{B}_m = gather(\mathbf{B}_e, \mathbf{B}_i^{rest}, \mathbf{B}_j^{rest})$
18: **end function**

---

**Algorithm 2** Runtime Compression

---

**Input:** query-passage token representations $\mathbf{h}_k$, compression ratio $r_p$
**Output:** merged text token representations $\mathbf{h}_n$

1: **for** each layer $\ell$ from layer $k + 1$ to $n$ in the reader encoder, **do**
2:    merge $\mathbf{h}_\ell$ using cosine distance $\mathcal{D}$: $\mathbf{h}_{\ell+1} = \text{bipartite\_merge}(\mathbf{h}_\ell, r_p, \mathcal{D})$;
3: **end for**                                              ▷ intra-passage token compression
4: **for** each layer $d$ from layer 1 to $n$ in the reader decoder, **do**
5:    **if** $d \% g == 0$ **then** continue **end if**            ▷ skip merging for $g$ layers
6:    merge $\mathbf{h}_\ell$ using cosine distance $\mathcal{D}$: $\mathbf{h}_{\ell+1} = \text{bipartite\_merge}(\mathbf{h}_\ell, r_p, \mathcal{D})$;
7: **end for**                             ▷ cross-passage token compression **return** $\mathbf{H}_n$

---

## A.2 IMPLEMENTATION DETAILS

**BTR Details.** We develop BTR based on the Atlas codebase using PyTorch 1.13.1 and HuggingFace Transformers v4.18.0 (Wolf et al., 2020). We conducted training using 4 to 8 A40 or A100 GPUs (depending on their availability on our cluster) with BF16 mixed precision. We list the training hyperparameters in Table 4. The decomposition layer $k$ is 9 for the BTR-Atlas base model, $k$ is 20 for the BTR-Atlas large model, and $k$ is 9 BTR-BERT. We implemented BTR-BERT using the original DPR (Karpukhin et al., 2020) codebase.

## A.3 BASELINE DETAILS.

**Atlas** (Izacard et al., 2022b) models obtain state-of-the-art performance on a wide range of knowledge-rich NLP tasks. Atlas model retrieves passages from the corpus using a Contriever (Izacard et al., 2022a)-based retriever, which is a BERT-like architecture with 110 million parameters shared between the query and the passage encoder. Since our focus is on the reader component, we use the retriever in the Atlas 11B model to obtain query-relevant passages and fine-tune Atlas reader models with these retrieved passages. Due to computing budget constraints, we compare the base (with 220M

Table 4: Training hyperparameters for BTR-Atlas.

| Hyperparameters | NQ base | NQ large | TQA base | TQA large | WQ base | WQ large | Fever base | Fever large | MMLU base | MMLU large |
|---|---|---|---|---|---|---|---|---|---|---|
| batch size | 8 | 4 | 8 | 4 | 8 | 4 | 8 | 4 | 4 | 2 |
| learning rate | 6e-5 | 6e-5 | 4e-5 | 4e-5 | 8e-5 | 8e-5 | 6e-5 | 6e-5 | 5e-5 | 5e-6 |
| training steps | 20000 | 20000 | 20000 | 20000 | 3000 | 3000 | 10000 | 10000 | 2000 | 2000 |
| warmup steps | 200 | 200 | 200 | 200 | 200 | 200 | 200 | 200 | 50 | 50 |
| weight decay | 0.01 | 0.01 | 0.01 | 0.01 | 0.01 | 0.01 | 0.01 | 0.01 | 0.01 | 0.01 |
| number of passages | 40 | 40 | 40 | 40 | 40 | 40 | 40 | 40 | 30 | 30 |
| max query length | 40 | 40 | 64 | 64 | 40 | 40 | 40 | 40 | 256 | 256 |
| max passage length | 320 | 320 | 320 | 320 | 320 | 320 | 320 | 320 | 320 | 320 |
| max answer length | 32 | 32 | 32 | 32 | 32 | 32 | 32 | 32 | 32 | 32 |

parameters) and large (770M parameters) size variants. We use the original hyperparameters to finetune Atlas.

**Atlas-Q** is a quantized version of Atlas. We use GTPQ (Frantar et al., 2023) to quantize the fine-tuned Atlas baseline model in 4bit. We do not directly use quantization-aware training such as QLoRA (Dettmers et al., 2023) since the quantized model's inference is much slower.

**DensePhrase** (Lee et al., 2021) creates dense representations of phrases from the passages. It retrieves answers from the phrase index without using the retrieve-and-read pipeline and achieves high query processing throughput. We run the official DensePhrase implementation on an A100 GPU and report the throughput numbers.

**DeFormer** (Cao et al., 2020) speeds up the inference in encoder-only reader models by decomposing the passage and query encoding and caching the continuous passage representations. We apply DeFormer over Atlas and fine-tune the adapted model. In fact, we can view DeFormer as a simplified system from BTR without binary representations and the offline and runtime compression components.

**LLaMA2** (Touvron et al., 2023a;b) consists of open-source large language models that achieve top performance on many NLP benchmarks including the knowledge tasks we study. Since the few-shot versions still lag behind fine-tuned models, we finetune the LLaMA2-7B model using the LoRA adapter and report its results on the evaluation tasks. Due to memory constraints, we limit the retrieved passages to 5. We implemented this baseline using the PEFT library (Mangrulkar et al., 2022), the LoRA config is as follows: r=8, alpha=32, dropout=0.05, task_type="CAUSAL_LM", target_modules=.*(q|k|v|o|gate|down|up)_proj.*, bias="none".

## A.4 DATASETS DETAILS

Table 5: Statistics of the number of examples for the evaluation datasets.

| | NQ | TQA | WQ | Fever | MMLU |
|---|---|---|---|---|---|
| Train | 79168 | 78785 | 3400 | 145449 | 95127 |
| Validation | 8757 | 8837 | 378 | 19998 | 1531 |
| Test | 3610 | 11313 | 2032 | 19998 | 14042 |

**Open-domain QA**. We use the original open-domain variants of NaturalQuestions (Kwiatkowski et al., 2019), TriviaQA (Joshi et al., 2017), WebQuestions (Berant et al., 2013). The NaturalQuestions (NQ) dataset consists of natural Google search questions that have answers from Wikipedia articles. The open-domain variant contains 91,535 question-answer pairs (79168 for training, 8757 for development, and 3610 for testing). TriviaQA (TQA) has 98,935 quiz-style trivia questions (78785 for training, 8837 for validation, 11313 for testing), its unfiltered version is used for open-domain QA. The questions from WebQuestions (WQ) are constructed from Freebase KB and Google Suggest API and their answers are labeled by Amazon Mechanical Turk. It has 3,778 examples for training and 2,032 for testing. We reserve 10% of the training set for development and use the rest for training.

**Fact Checking**. We use the original FEVER (Thorne et al., 2018) dataset for the fact-checking task. FEVER contains 145k training examples, and both development and testing tests have 20k examples.

Each example has a claim sentence and a 3-class (supported, refuted, or not enough info) label. The task requires retrieving evidence from the Wikipedia corpus and making a judgment.

Table 6: Inference throughput numbers for the baselines and BTR-Atlas.

|                  | NQ   | TQA  | WQ   | FEVER | MMLU |
|------------------|------|------|------|-------|------|
| DeFormer         | 22.4 | 16.7 | 18.8 | 24.6  | 9.6  |
| DensePhrase      | 50.4 | 48.5 | 48.1 | -     | -    |
| Atlas-Quant      | 12.5 | 13.2 | 11.2 | 11.8  | 6.4  |
| LLaMA-7B         | 1.2  | 0.8  | 1.3  | 1.2   | 0.5  |
| Atlas base       | 10.2 | 8.4  | 9.5  | 10.5  | 5.2  |
| **BTR-Atlas** base  | 31.5 | 21.4 | 25.1 | 32.1  | 13.4 |
| Atlas large      | 3.8  | 2.1  | 2.2  | 3.5   | 3.3  |
| **BTR-Atlas** large | 15.3 | 8.2  | 7.9  | 14.3  | 8.1  |

**Knowledge-Intensive Reasoning**. We use the mass-multitask language understanding (MMLU) benchmark (Hendrycks et al., 2020) to evaluate the knowledge reasoning ability of RaLMs. MMLU is a multiple-choice question-answering dataset that has 57 subdatasets covering various domains like history, math, law, etc. The benchmark tests models' world knowledge and problem-solving ability. The training set has 95127 examples, the development set has 1531 examples, and the test set has 14042 examples.

Table 5 summarizes the number of examples for each split in the datasets.

## A.5 BASELINE RESULTS

Table 6 contains the inference throughput numbers of the baselines and BTR-Atlas models. Table 7 presents the same inference and accuracy numbers as in Table 1 but with added standard deviation numbers.

## A.6 ABLATION RESULTS

Table 8 presents the accuracy and speedup results of Atlas base and BTR-Atlas base models using different numbers of passages on the NQ dataset.

Table 9 includes the detailed numbers of the compression ratio ablation experiments.

Table 7: Task performance and inference efficiency comparison between the baseline models and BTR-Atlas over five evaluation tasks. '±' denotes the standard deviation of accuracy or em scores across five runs with different random seeds. The speedup (Sp) deviations are quite small (<0.5%) which we omit here.

| | NQ | | TQA | | WQ | | FEVER | | MMLU | |
|---|---|---|---|---|---|---|---|---|---|---|
| | EM | Sp | EM | Sp | EM | Sp | Acc | Sp | Acc | Sp |
| Atlas base | $52.1_{\pm0.3}$ | 1.0 | $69.3_{\pm0.4}$ | 1.0 | $46.4_{\pm0.4}$ | 1.0 | $72.9_{\pm0.2}$ | 1.0 | $38.6_{\pm0.5}$ | 1.0 |
| Atlas-Q | $51.8_{\pm0.1}$ | 1.2 | $68.5_{\pm0.3}$ | 1.3 | $45.1_{\pm0.2}$ | 1.1 | $70.4_{\pm0.3}$ | 1.1 | $37.8_{\pm0.3}$ | 1.2 |
| DeFormer | $51.4_{\pm0.2}$ | 2.2 | $68.0_{\pm0.4}$ | 2.0 | $44.8_{\pm0.5}$ | 2.0 | $71.8_{\pm0.5}$ | 2.3 | $33.9_{\pm0.4}$ | 1.8 |
| DensePhrase | $40.9_{\pm0.2}$ | 4.9 | $53.6_{\pm0.4}$ | 5.8 | $37.5_{\pm0.5}$ | 5.1 | - | - | - | - |
| LLAMA2-7B | $47.8_{\pm0.2}$ | 0.1 | $74.3_{\pm0.3}$ | 0.1 | $51.2_{\pm0.3}$ | 0.1 | $76.3_{\pm0.2}$ | 0.1 | $51.2_{\pm0.4}$ | 0.1 |
| **BTR**-Atlas base | $49.5_{\pm0.3}$ | 3.1 | $66.7_{\pm0.2}$ | 2.5 | $43.8_{\pm0.4}$ | 2.6 | $70.2_{\pm0.5}$ | 3.1 | $35.4_{\pm0.4}$ | 2.6 |
| Atlas large | $58.3_{\pm0.3}$ | 1.0 | $73.6_{\pm0.2}$ | 1.0 | $51.5_{\pm0.5}$ | 1.0 | $78.2_{\pm0.4}$ | 1.0 | $41.1_{\pm0.6}$ | 1.0 |
| **BTR**-Atlas large | $56.1_{\pm0.4}$ | 4.0 | $70.8_{\pm0.3}$ | 3.9 | $49.1_{\pm0.5}$ | 3.6 | $75.9_{\pm0.4}$ | 4.1 | $39.2_{\pm0.6}$ | 2.5 |

| #of passages | Atlas base EM | BTR-Atlas base EM | Speedup |
|---|---|---|---|
| 5 | 41.1 | 38.9 | 2.9 |
| 10 | 43.5 | 41.9 | 3.2 |
| 20 | 46.1 | 43.8 | 3.4 |
| 30 | 47.2 | 45.4 | 3.3 |
| 40 | 52.1 | 49.5 | 3.1 |

Table 8: Accuracy and speedup comparison between Atlas base and BTR-Atlas base for different numbers of retrieved passages on the NQ dataset.

| Compression Ratio | Storage (GB) | Accuracy (%) |
|---|---|---|
| 0.05 | 151 | 50.2 |
| 0.1 | 143 | 49.8 |
| 0.15 | 135 | 49.6 |
| 0.2 | 127 | 49.5 |
| 0.25 | 119 | 48.2 |
| 0.3 | 111 | 47.1 |
| 0.35 | 103 | 46.2 |
| 0.4 | 95 | 43.5 |
| 0.45 | 87 | 38.4 |
| 0.5 | 80 | 32.1 |

(a) Accuracy and storage effects for different offline token compression ratios.

| Compression Ratio | Accuracy (%) | Throughput (QPS) |
|---|---|---|
| 0.05 | 50.1 | 28.5 |
| 0.1 | 49.8 | 29.2 |
| 0.15 | 49.6 | 30.4 |
| 0.2 | 49.5 | 31.5 |
| 0.25 | 48.7 | 32.2 |
| 0.3 | 47.6 | 34.4 |
| 0.35 | 45.2 | 35.6 |
| 0.4 | 43.2 | 37.5 |
| 0.45 | 40.6 | 38.9 |
| 0.5 | 38.1 | 41.2 |

(b) Accuracy and throughput effects for different intra- and cross- passage token compression ratios.

Table 9: Accuracy, storage and throughput comparisons for different two-stage token compression ratios on the NaturalQuestions dataset. To achieve a good balance between accuracy, storage and throughput in BTR, we choose compression ratio 0.2 for both online and offline token compression.

