# OpenReview forum: "BTR: Binary Token Representations for Efficient Retrieval Augmented Language Models"
_ICLR.cc/2024/Conference — ICLR 2024 spotlight_

### Official Review · Reviewer_PDbq · 2023-10-28

**Soundness:** 3 good
**Presentation:** 4 excellent
**Contribution:** 3 good
**Rating:** 6
**Confidence:** 4

**Summary:**

This article introduces Binary Token Representation (BTR) to solve the problem that running retrieval-enhanced language models (LM) is slow and difficult to scale. Due to the potential loss of accuracy, the authors propose a passage representation recovery objective and a query-aware passage token distillation objective to restore performance. By combining offline and runtime compression, the proposed method speeds up state-of-the-art inference by 4x and reduces storage space by more than 100x in five knowledge-intensive NLP tasks, while maintaining over 95% of task performance.

**Strengths:**

1.  For the first time, the authors construct binary token representations to improve the efficiency of retrieval augmentation models, an approach that has never been explored before.

2. The paper is very well-written and well-organized and provides clear motivation, background, and some technical details for the proposed model, including model quantization and binarization.

3.  Experiments are conducted on 5 datasets and provide meaningful comparisons with existing retrieval-enhanced language models such as Atlas and DensePhrase and large language models such as LLaMA2-7B. The results show that the proposed model can maintain 95% of the above task performance while improving the inference speed and greatly reducing the storage space. The paper also includes ablation studies to analyze the performance further.

**Weaknesses:**

1. As the paper mentions, BTR is difficult to apply to decoder-only models, and most current large language models utilize a decoder-only structure.

2. The paper only discusses FiD, which is the SOTA model in the KiLT ranking. However, it is also important to discuss other commonly used retrieval-augmented structures, such as RAG, particularly when dealing with black-box LLM.

3. For retrieval augmented language models, the number of retrieved passages is an important factor for performance and efficiency. While
authors only set this factor to a fixed number (40 or 30 for different datasets), thus additional experiments should be involved.

**Questions:**

**A**: In addition to BERT, what other experiments have you conducted on the Encoder-Only model? For example, Deberta, COCO-LM, and so on.

**B**: The author claims to have utilized different random number seeds for 5 runs. It would be beneficial to include the standard deviation of the results in Table 1.

**C**: In the Calibrated binarization section (Section 3.1), the author asserts that employing a straight-through estimator (STE) yields superior results compared to combining the annealing method with the tanh function. Are there any analytical experiments conducted to substantiate this claim? Additionally, is it viable to utilize a linear layer and sigmoid function for binarization? I recommend the author to include additional binarization methods in the experimental section to leverage the benefits of STE.

---

> ### Author Response · Authors · 2023-11-17
>
> We thank the reviewer for acknowleding our binary token representations method is new and unexplored before. We highlight the answers to the following questions:
>
> >As the paper mentions, BTR is difficult to apply to decoder-only models, and most current large language models utilize a decoder-only structure.
>
> We acknowledge that BTR in its current form does not apply to decoder-only models, we believe extending BTR for decoder-only models will be an important future work. For example, one could explore using the BTR method to compress the KV-cache to speed up decoder models during inference.
>
> >The paper only discusses FiD, which is the SOTA model in the KiLT ranking. However, it is also important to discuss other commonly used retrieval-augmented structures, such as RAG, particularly when dealing with black-box LLM.
>
> Since we focus on improving the reader efficiency of retrieval-augmented models, it will be difficult to optimize the reader of black-box RAG models if we do not know the LLM model structure. However, it will be useful to extend BTR to make it work with decoder-only LLMs.
>
> >For retrieval augmented language models, the number of retrieved passages is an important factor for performance and efficiency. While authors only set this factor to a fixed number (40 or 30 for different datasets), thus additional experiments should be involved.
>
> We followed the practice of the state-of-the-art retrieval augmented language model called Atlas. Atlas set 40 passages for the tasks by default. In some cases, like the MMLU dataset, it uses 30 passages because more passages saturate the task performance. Setting different numbers of retrieved passages will affect original Atlas reader efficiency and performance but is orthogonal to our BTR method, i.e. if more passages boost Atlas accuracy, our BTR accuracy will also improve and the efficiency gains will be similar.
>
> >A: In addition to BERT, what other experiments have you conducted on the Encoder-Only model? For example, Deberta, COCO-LM, and so on.
>
> We haven’t conducted other encoder-only experiments, we only experimented with the BERT model to showcase that our BTR method does apply to encoder-only models. We believe that BTR will also apply to other encoder-only models by applying similar procedures like BERT.
>
> >B: The author claims to have utilized different random number seeds for 5 runs. It would be beneficial to include the standard deviation of the results in Table 1.
>
> We have included the deviation number in the appendix since adding them to Table 1 makes the table very busy.
>
> >C: In the Calibrated binarization section (Section 3.1), the author asserts that employing a straight-through estimator (STE) yields superior results compared to combining the annealing method with the tanh function. Are there any analytical experiments conducted to substantiate this claim? Additionally, is it viable to utilize a linear layer and sigmoid function for binarization? I recommend the author to include additional binarization methods in the experimental section to leverage the benefits of STE.
>
> The sigmoid function has a slower rate of change than the tanh function and its derivative will be smaller than the tanh derivative, therefore using the sigmoid function for binarization could lead to slower convergence. We built our calibrated binarization idea on the HashNet method (Cao et al, 2017) that uses tanh function to implement the binarization. We leave exploring other binarization techniques to future work.

---

> > ### Comment · Reviewer_PDbq · 2023-11-21
> >
> > Thanks to the authors for the detailed response, and most of my concerns are addressed.
> > However, I suggest that the experiment on a different number of documents is important in this paper, while simply saying "our BTR accuracy will also improve and the efficiency gains will be similar" is not valid and strong evidence for the readers.
> > Also, the experiments on other encode-only models also have the same problem, and I prefer to see the results in the revised version, instead of "we believe ...".
> > Thus, I keep my positive score.

---

> > > ### Author Response · Authors · 2023-11-22
> > >
> > > Thanks for the comments, we were able to get the results for different numbers of passages on the NQ dataset, the storage benefits remain the same and we show the accuracy (EM) and speedup below:
> > >
> > > | number of passages   | Atlas base EM | BTR-Atlas base EM | Speedup |
> > > |----|---------------|-------------------|---------|
> > > |  5 |          41.1 |              38.9 |     2.9 |
> > > | 10 |          43.5 |              41.9 |     3.2 |
> > > | 20 |          46.1 |              43.8 |     3.4 |
> > > | 30 |          47.2 |              45.4 |     3.3 |
> > > | 40 |          52.1 |              49.5 |     3.1 |
> > >
> > >
> > > Due to time constraints, we are still working on adding the experiments for using a different encoder model. We will add the results to the revised paper.

---

### Official Review · Reviewer_KKL3 · 2023-10-31

**Soundness:** 3 good
**Presentation:** 3 good
**Contribution:** 4 excellent
**Rating:** 8
**Confidence:** 4

**Summary:**

The authors introduce multiple compression techniques to speed up retrieval-augmented models collectively referred to as BTR. These techniques are specific to a certain class of models like BERT and Atlas. And it's not clear that they will generalize to other types of retrieval-augmented LMs. There are many steps in the compression that are all required for good speed and accuracy. These are covered well in the ablation table:

* binary passage representation is needed to reduce storage (speed is almost the same when using the dense passages).
* offline compression gives a minor improvement in storage and speed at cost of accuracy.
* online compression gives big speed improvements.
* passage token distillation improves representation of dense vectors, and is important for good accuracy. This filters for 50 percent of the most salient tokens according to attention, although it is not clear if this was necessary and maybe distillation of 50 percent random tokens would yield similar performance.
* passage recovery loss for the binary vector retains semantic information of the dense vector, and is important for good accuracy.

The results empirically show the speed (2-4x) and storage benefits (100x) of BTR, albeit at about 5 percent drop in performance. The paper is clearly very informative with analysis and details, but sometimes some basic details are missing and the benefits of BTR are occasionally oversold.

**Strengths:**

1. Storage savings of 100x. This will make retrieval more available to many, since storage can often become a bottleneck in retrieval-augmented ML.

2. Speed savings of 2-4x. Although this is similar speed improvement as Deformer (Cao et al), it leads to 5 percent accuracy loss which is worse than Deformer's 1 percent accuracy loss.

3. Extensive experiments and analysis. There are some confusing or missing details, but these can be probably be easily fixed.

**Weaknesses:**

1. There are many steps required to make BTR work well. Of course, that is also why this is a valuable paper since it outlines what these steps are.

2. There is a substantial (about 5 percent) drop in model accuracy. It's not clear whether 2-4x speed boost is enough value to make up for this, although the storage improvements are definitely very valuable.

3. The paper is not very self-contained. It seems like the reader is expected to have read Cao et al and Atlas papers very closely. This is not ideal. For example, it is hard to understand what it means by "decomposed model" or "decomposed reader" in sec 3.2. Also, the reader is left to infer that retrieval is done at the passage level, but passages are incorporated at the token-level.

4. (minor) The paper is hard to read. In more than one instance, a method is used but is "defined below", so we must constantly revisit parts of the reading to get a full understanding.

5. (medium) "BTR is more accurate and efficient than other baseline methods" This claim can easily be interpreted as "all other baselines", which is not accurate. There are baselines that are more accurate or more efficient. This should probably be revised to be more accurate/specific. Also, I am not sure why BTR base is constantly bolded in Sp column when it is not the best value.

6. There is not a good breakdown of efficiency between retrieval + inference. Perhaps alternative methods akin to DistilBERT will give a substantial speed boost plus keep good performance.

7. (minor) Similarly, given the emphasis on storage it would have been helpful to see some basic baselines to improve storage. Although I am not sure what else can easily achieve 100x savings without larger tradeoff in performance.

8. (very minor) It was confusing whether the token distillation is taken directly from Cao et al, or is something new.

**Questions:**

Q1: Why do we need Step 2 for the decomposed reader? Doesn't Step 1 already provide a reader?

Q2: "This is likely because prior work only considers a single passage where all tokens are important to the task" What does this mean? Surely many prior works use more than one passage.

typo: Funsion-in-Decoder

---

> ### Author Response · Authors · 2023-11-17
>
> We thank the reviewer for recognizing the significant storage and speed savings brought by our method. We provide the following responses for the reviewer’s questions below:
>
> > There is a substantial (about 5 percent) drop in model accuracy. It's not clear whether 2-4x speed boost is enough value to make up for this, although the storage improvements are definitely very valuable.
>
> We want to clarify that the accuracy drop for the studied tasks is around 2\~3 absolute percentages. The relative percentage drop can be 5 percent because dividing the 2\~3 absolute drop by a relatively smaller task accuracy number will make the relative percentage number higher.
>
>
> > The paper is not very self-contained. It seems like the reader is expected to have read Cao et al and Atlas papers very closely. This is not ideal. For example, it is hard to understand what it means by "decomposed model" or "decomposed reader" in sec 3.2. Also, the reader is left to infer that retrieval is done at the passage level, but passages are incorporated at the token-level.
>
> Thanks for making the writing suggestions. We have revised the paper based on your questions; specifically, 1) we clarified what we mean by decomposition (Section 3.2) .The decomposed reader refers to the model without precomputing the passage representations in the reader encoder model. We use decomposition to denote the process where we break down the passage and query encoding in the encoder's lower layers, Figure 2 illustrates how the decomposition enables passage precomputation.  2) We also revised the background section to reflect passage-level retrieval versus our token-level passage representation for readers.
>
> >(minor) The paper is hard to read. In more than one instance, a method is used but is "defined below", so we must constantly revisit parts of the reading to get a full understanding.
>
> For easier reading, we will improve the writing with fewer forward references. But could you help clarify which parts are not clear? Do you mean the terms or methods in section 3 overview paragraph?
>
> >(medium) "BTR is more accurate and efficient than other baseline methods" This claim can easily be interpreted as "all other baselines", which is not accurate. There are baselines that are more accurate or more efficient. This should probably be revised to be more accurate/specific. Also, I am not sure why BTR base is constantly bolded in Sp column when it is not the best value.
>
> We agree that the wording was not great in the submission. We revised the wording in the paper to clarify the benefits of BTR (section 4.3). BTR provides a favorable tradeoff because it has larger efficiency benefits while maintaining task accuracy. Our focus in this work is to provide faster inference speed and use less storage while maintaining task accuracy for retrieval-augmented language models. We fixed the Sp column style to be consistent.
>
> >There is not a good breakdown of efficiency between retrieval + inference. Perhaps alternative methods akin to DistilBERT will give a substantial speed boost plus keep good performance.
>
> We focus on improving the reader efficiency in this work and studying retrieval efficiency will be an interesting and important future work. We haven’t used models like DistillBERT to compare the efficiency and task performance,  in this work, we mostly focus on improving storage, whereas DistillBERT only focuses on compression. But our BTR method is orthogonal to using smaller models and can be combined to models like DistillBERT to improve storage efficiency. In fact, one can apply BTR to smaller models to further boost efficiency.
>
>
> >(very minor) It was confusing whether the token distillation is taken directly from Cao et al, or is something new.
>
> We designed a new passage token distillation that is query-aware and different from Cao et al.
>
> >Q1: Why do we need Step 2 for the decomposed reader? Doesn't Step 1 already provide a reader?
>
> There are two sources of information loss, decomposition and binarization. Step2 decomposed reader without binarization has the same decomposed model architecture as the step3 model, it provides a better teacher for the step3 model to learn. If we directly use the step1 reader as the teacher for the step3, the passage information will not be effectively distilled into step3 reader because their model architectures are different.
>
> >Q2: "This is likely because prior work only considers a single passage where all tokens are important to the task" What does this mean? Surely many prior works use more than one passage.
>
> Thanks for pointing out the ambiguity here, we specifically discuss the distillation in prior work DeFormer (Cao et al) that distills all tokens for single passage scenarios. Other prior works like open-domain question answering do use multiple passages but do not use distillation. We clarified this in the revised paper.

---

> > ### Comment · Reviewer_KKL3 · 2023-11-22
> >
> > Thank you for the effort in the rebuttal. I believe with the new edits the writing is substantially more clear.
> >
> > There are three places where "defined below" happens. It is most egregious in 3.2 since although "query-aware passage token distillation" and "passage representation recovery" are somewhat intuitive names, they can have multiple interpretations and the section only really makes sense to me if I read it twice. As I mentioned, this is a minor complaint and perhaps a personal style preference by me. If you are looking for suggestions, then it could be nice to provide more intuition behind what both of these things are doing in Step 2 and 3, so the reader does not need to guess from the name.

---

### Official Review · Reviewer_U3RY · 2023-11-01

**Soundness:** 2 fair
**Presentation:** 3 good
**Contribution:** 3 good
**Rating:** 6
**Confidence:** 3

**Summary:**

This paper proposes binary token representations for retrieval-augmented LMs.
The goal is to increase inference speed and reduce storage requirements.
The key idea to achieve this in BTR is to create cacheable binary token representations of the retrieved passages such that the passage encoding can be precomputed offline and stored in a compact format.
Empirical evaluation on knowledge-intensive NLP tasks shows the effectiveness of their approach w.r.t. corresponding non-optimized retrieval-augmented LM:
* inference speed: 2-4x
* storage: up to 100x reduction (e.g., 10TB --> 100GB)
* task performance: retains 90-95% of original performance

**Strengths:**

* Importance and relevance of topic: LLMs are everywhere and retrieval augmentation LMs addresses critical problems in LLMs such as hallucination, staleness, and privacy leaks, but suffer from low inference speed and huge storage requirements.
* BTR has much lower storage footprint than other approaches, is more scalable and has lower inference speed at the expense of a "modest" loss in performance.
* The paper reports numbers for actual throughput for a more realistic comparison of efficiency across different systems.
* The code will be publicly available.

**Weaknesses:**

* "BTR is more accurate and efficient than other baseline methods.": According to Figure 3, the proposed BTR appears to be on the Pareto front. I.e., it isn't substantially better than existing methods but provides a different tradeoff between speed and accuracy. This makes it hard to assess the merits of BTR. To make the result less sensitive to a specific operating point and better comparability, it would be interesting to show, for example, how the tradeoff changes with the resolution of the representation (1-bit vs b-bit representations). Furthermore, adding sub-optimal points to the plots would give a more comprehensive picture (for example, LLaMA at different size/speed/accuracy).
* The benefits at runtime come with a clearly more complicated training pipeline and increased training time. How much?
* Figure 3: Why do the plots include Atlas-Q but not the original model (Atlas)? Also, I can't see any small points for Atlas-Q and LLaMA2-7B.
* Different spelling in title and text: "retrieval augmented language model" vs "retrieval-augmented language model".
* Funsion-in-Decoder → Fusion-in-Decoder

**Questions:**

* The text accompanying Figure 3 says "BTR presents better efficiency versus accuracy trade-offs by maintaining high
accuracy and inference throughput with a smaller storage footprint.": Could you please clarify what you mean with this? As for me, BTR is another point on the Pareto front.
* A method that allows to choose an operating point on the Pareto front would be more useful in practice than a method for a single operating point. Can BTR be extended along this dimension?
* Could you please say a few words on the complexity of the training pipeline?

---

> ### Author Response · Authors · 2023-11-17
>
> Thank you for acknowledging the importance of improving the efficiency of retrieval LMs. We have revised the paper following your clarification requests and answer your questions/comments below:
>
>
> >The text accompanying Figure 3 says "BTR presents better efficiency versus accuracy trade-offs by maintaining high accuracy and inference throughput with a smaller storage footprint.": Could you please clarify what you mean with this? As for me, BTR is another point on the Pareto front.
>
>
> We revised the paper to define the better efficiency trade offs under certain a accuracy target (e.g, afford relatively 5% accuracy drop). A model has a better trade-off if the model is more efficient (less storage and faster speed) when satisfying the accuracy budget.
>
> We added a Pareto curve in Figure 3, BTR is indeed a point on the Pareto front. BTR lies in the center of the Pareto curve, in general, the top right smaller circle is better. Compared to other neighboring points on the Pareto curve, BTR provides more efficiency benefits and maintains accuracy; for example, BTR is fast and has much less storage than DeFormer with comparable accuracy,  DensePhrase is faster than BTR but comes with very significant accuracy drops ( >10 absolute points) and BTR is more storage-efficient.
>
>
>
> >how the tradeoff changes with the resolution of the representation (1-bit vs b-bit representations). Furthermore, adding sub-optimal points to the plots would give a more comprehensive picture (for example, LLaMA at different size/speed/accuracy).
> A method that allows to choose an operating point on the Pareto front would be more useful in practice than a method for a single operating point. Can BTR be extended along this dimension?
>
> We acknowledge that adding more operating points would be useful to understand the tradeoffs. However, we want to clarify that we do not aim to propose a new quantization method that can quantize/compress model parameters into different bit widths (e.g. 2-bit or 4-bit). Instead, our method compresses the passage token representations into 1-bit binary vectors, i.e., compression model activations. Extending BTR to b-bit b-bit (e.g. 2-bit or 4-bit) requires other quantization techniques that are not directly comparable to our 1-bit binarization method. However, BTR can provide more operating points by adjusting the decomposition layer, offline, and runtime compression ratios in the step 2 and step 3 training. For example, one can set a relatively lower decomposition layer (smaller k) but set a higher runtime compression ratio to provide a similar speedup as our setting. Intuitively, setting a higher decomposition layer (larger k) or a larger compression ratio will provide bigger speedups but come with a bigger accuracy loss.
>
>
> >Figure 3: Why do the plots include Atlas-Q but not the original model (Atlas)? Also, I can't see any small points for Atlas-Q and LLaMA2-7B.
>
> We updated the figure and added the original Atlas results. Initially, we didn’t plot the Atlas model baseline because it has very similar accuracy and efficiency data points as the Atlas-Q. The data points for Atlas-Q and LLaMA2-7B are very small because their storage is 0 whereas other systems have storage usage, we updated the figure for easier display (the figures are vector images, so one can zoom in to see the small points for Atlas-Q and LLaMA2-7B).

---

> > ### Comment · Reviewer_U3RY · 2023-11-22
> >
> > Thanks to the authors for the clarifications.
> >
> > > We revised the paper to define the better efficiency trade offs under certain a accuracy target (e.g, afford relatively 5% accuracy drop). A model has a better trade-off if the model is more efficient (less storage and faster speed) when satisfying the accuracy budget.
> >
> > Fair enough, although 5% accuracy drop feel a bit arbitrary and the reference points from the literature probably were optimized toward a different efficiency target. But don't let that distract us from the main contribution of BTR's significantly improved storage efficiency.
> >
> > > For example, one can set a relatively lower decomposition layer (smaller k) but set a higher runtime compression ratio to provide a similar speedup as our setting. Intuitively, setting a higher decomposition layer (larger k) or a larger compression ratio will provide bigger speedups but come with a bigger accuracy loss.
> > In case I missed it in the paper, could you please add such a tradeoff curve to the paper.
> >
> > Overall, I stick with my rating as "marginal accept".

---

### Official Review · Reviewer_X7Hc · 2023-11-02

**Soundness:** 3 good
**Presentation:** 3 good
**Contribution:** 3 good
**Rating:** 6
**Confidence:** 3

**Summary:**

This paper presents a method to accelerate inference speed of retrieval augment language models, while reducing the required storage space. The authors use binary token representations and compression (for collapsing the embeddings of similar tokens) to increase speed. They show that their method maintains up to 95% task performance when compared with the base method.

**Strengths:**

The strengths are as follows:
* Retrieval augment language models are becoming increasingly popular. Increasing the inference speed and reducing the memory footprint of such methods will be quite useful. The authors demonstrate that they can achieve a 4x speedup in inference speed and a 100x reduction in storage space.
* The authors present results across multiple datasets.
* The authors ablate all of their modifications and show how each one affects performance, speed and memory.

**Weaknesses:**

The weaknesses are as follows:
* Some things like passage representation regularization are mentioned in passing and it would be helpful if the authors added a couple sentences providing context and explaining what this is.
* No motivation or explanation is given for why distillation is required/helpful.
* It is not clear if the linear projection layer for passage representation recovery is a learnable layer. Also, the $L_{recovery}$ term is confusing; $b_i$ is the binary passage representation and $h_i$ is the original passage representation, where is the projection layer used?
* No comparison to newer methods like Lumen.

**Questions:**

* Did you try different thresholds for r%? How much did the performance change?
* Is it important to do runtime compression after every block?
* How slow is the bipartite matching for compression?

**Details Of Ethics Concerns:**

No ethics review required.

---

> ### Author Response · Authors · 2023-11-17
>
> Thank you for supporting our paper and acknowledging our method is useful for increasing the efficiency of retrieval augment language models. We add clarifications to the questions below:
>
> >Some things like passage representation regularization are mentioned in passing and it would be helpful if the authors added a couple sentences providing context and explaining what this is.
>
> Thanks for the clarification request. Some details were missed due to lack of space. We have currently revised the paper (section 3.2). We adopt passage representation distillation from DeFormer (Cao et al.,2020) to address the passage information loss issue due to decomposition, because passage representations in the lower Transformer layers no longer depend on the question. Distilling upper-layer passage representations from the original (not decomposed) reader encoder helps reduce such information loss and improves task performance.
>
> >No motivation or explanation is given for why distillation is required/helpful.
>
> Distilling passage representations, especially for the upper layers in the reader encoder, helps the decomposed reader model produce passage representations that carry similar information as the corresponding layers in the original (not decomposed) model. This helps reduce the task performance loss due to the lack of full query-passage interactions in the encoder's lower layers. Our query-aware passage token distillation helps pass more fine-grained query-related information from the original passage representations to decomposed passage representations. We revised the writing (in section 3.2) accordingly.
>
> >It is not clear if the linear projection layer for passage representation recovery is a learnable layer. Also, the $L_{recovery}$ term is confusing; $b_i$ is the binary passage representation and $h_i$ is the original passage representation, where is the projection layer used?
>
> We add clarification to section 3.2. The linear projection layer is a learnable layer (parameterized by $\mathbf{W}\_{proj}$ and $\mathbf{bias}\_{proj}$) and we apply the projection layer over the binary representations, $\mathcal{L}\_{recovery} = \frac{1}{d}\sum\_{i=1}^d (h\_i - b^{proj}\_i)^2$, where $\mathbf{b}^{proj}=\mathbf{W}\_{proj}\mathbf{b}+\mathbf{bias}\_{proj}$.
>
> >No comparison to newer methods like Lumen.
>
> We cited Lumen method in the related work. Unfortunately, the source code for the Lumen has not been released, making the empirical evaluation difficult. However, as we described in the related work section, Lumen shares a similar idea to DeFormer, both precompute the passage representations in continuous float-point vectors that use 100x more storage than our BTR method.
>
> >Did you try different thresholds for r%? How much did the performance change?
>
> Yes, we study the threshold effect in section 4.4, Figure 4 presents the performance (accuracy) versus efficiency (storage for offline compression and throughput for runtime compression) tradeoffs.
>
>
> >Is it important to do runtime compression after every block?
>
> For the runtime compression that happens in the encoder k+1 and following layers, we apply the token merging for every layer/block which provides more inference speed benefits since there are not enough layers left (e.g. k is set to a relatively larger number 9 for a 12-layer encoder to avoid more passage computation). For decoder runtime compression, we apply it every few (e.g., 3 for a 12-layer decoder) layers because this provides a better empirical task performance versus inference speedup tradeoffs, setting compression after every block/layer can provide similar inference speedup but requires more tuning of the compression ratios/thresholds.
>
> >How slow is the bipartite matching for compression?
>
> We empirically do not notice too much (2\~5%, depending on the number of passage tokens for different tasks) the extra overhead of bipartite matching. The reduced tokens (merged after bipartite matching) enable a much faster (15\~25%) inference speed (table 3).

---

### Author Response · Authors · 2023-11-17
**General response**

We thank all the reviewers for their thoughtful comments and suggestions. We thank the reviewers for acknowledging that improving the efficiency of retrieval-augmented language models is an important research topic and our BTR method makes a significant contribution to reducing storage requirements and improving inference speed.

Reviewer U3RY and KKL3 have a comment on the complexity of our BTR training pipelines.
We empirically notice the training time is roughly 3x than the original pipeline since our BTR has 3 training steps and each takes about the same iterations/epochs as the original baseline (Atlas). We believe that the inference efficiency will amortize the extra training costs if our system gets deployed to a large volume setting.

We respond to each reviewer’s comments/questions below and we have also revised the paper accordingly (highlighted in blue in the paper).

---

### Meta-Review · Area_Chair_vBEM · 2023-12-20

**Metareview:**

This article introduces Binary Token Representation (BTR) to accelerate inference speed of retrieval augment language models, while reducing the required storage space.

Strength:
The proposed BTR method achieves 4x speed-up, and 100x in storage savings.

Weakness:
Inadequate explanation about passage representation regularization, justification of distillation, justification of drop in performance and gain in speedup.

**Justification For Why Not Higher Score:**

The method achieves certain tradeoff between performance drop and speedup gain. There still remain some doubt about justification of such tradeoff.

**Justification For Why Not Lower Score:**

All reviewers like the improvement over speedup and significant storage saving.

---

### Decision · Program_Chairs · 2024-01-16

Accept (spotlight)